# Auto-Conditioned Recurrent Networks for Extended Complex Human Motion Synthesis

**Yi Zhou[1]** [*]
zhou859@usc.edu

**Zimo Li[1]** [*]
zimoli@usc.edu

**Shuangjiu Xiao[2]**
xsjiu99@cs.sjtu.edu.cn

**Chong He[2]**
sal@sjtu.edu.cn

**Zeng Huang[1]**
zenghuan@usc.edu

**Hao Li[1,3,4]**
hao@hao-li.com

[1]University of Southern California    [2]Shanghai Jiao Tong University
[3]USC Institute for Creative Technologies    [4]Pinscreen

## Abstract

We present a real-time method for synthesizing highly complex human motions using a novel training regime we call the auto-conditioned Recurrent Neural Network (acRNN). Recently, researchers have attempted to synthesize new motion by using autoregressive techniques, but existing methods tend to freeze or diverge after a couple of seconds due to an accumulation of errors that are fed back into the network. Furthermore, such methods have only been shown to be reliable for relatively simple human motions, such as walking or running. In contrast, our approach can synthesize arbitrary motions with highly complex styles, including dances or martial arts in addition to locomotion. The acRNN is able to accomplish this by explicitly accommodating for autoregressive noise accumulation during training. Our work is the first to our knowledge that demonstrates the ability to generate over 18,000 continuous frames (300 seconds) of new complex human motion w.r.t. different styles.

## 1 Introduction

The synthesis of realistic human motion has recently seen increased interest (Holden et al., 2016; 2017; Fragkiadaki et al., 2015; Jain et al., 2016; Bütepage et al., 2017; Martinez et al., 2017) with applications beyond animation and video games. The simulation of human looking virtual agents is likely to become mainstream with the dramatic advancement of Artificial Intelligence and the democratization of Virtual Reality. A challenge for human motion synthesis is to automatically generate new variations of motions while preserving a certain style, e.g., generating large numbers of different Bollywood dances for hundreds of characters in an animated scene of an Indian party. Aided by the availability of large human-motion capture databases, many database-driven frameworks have been employed to this end, including motion graphs (Kovar et al., 2002; Safonova & Hodgins, 2007; Min & Chai, 2012), as well as linear (Safonova et al., 2004; Chai & Hodgins, 2005; Tautges et al., 2011) and kernel methods (Mukai, 2011; Park et al., 2002; Levine et al., 2012; Grochow et al., 2004; Moeslund et al., 2006; Wang et al., 2008), which blend key-frame motions from a database. It is hard for these methods, however, to add new variations to existing motions in the database while keeping the style consistent. This is especially true for motions with a complex style such as dancing and martial arts. More recently, with the rapid development in deep learning, people have started to use neural networks to accomplish this task (Holden et al., 2017; 2016; 2015). These works have shown promising results, demonstrating the ability of using high-level parameters (such as a walking-path) to synthesize locomotion tasks such as jumping, running, walking, balancing, etc. These networks do not generate new variations of complex motion, however, being instead limited to specific use cases.

---

[*]equal contribution

In contrast, our paper provides a robust framework that can synthesize highly complex human motion variations of arbitrary styles, such as dancing and martial arts, without querying a database. We achieve this by using a novel deep auto-conditioned RNN (acRNN) network architecture.

Recurrent neural networks are autoregressive deep learning frameworks which seek to predict sequences of data similar to a training distribution. Such a framework is intuitive to apply to human motion, which can be naturally modeled as a time series of skeletal joint positions. We are not the first to leverage RNNs for this task (Fragkiadaki et al., 2015; Jain et al., 2016; Bütepage et al., 2017; Martinez et al., 2017), and these works produce reasonably realistic output at a number of tasks such as sitting, talking, smoking, etc. However, these existing methods also have a critical drawback: the motion becomes unrealistic within a couple of seconds and is unable to recover.

This issue is commonly attributed to error accumulation due to feeding network output back into itself (Holden et al., 2017). This is reasonable, as the network during training is given ground-truth input sequences to condition its subsequent guess, but at run time, must condition this guess on its own output. As the output distribution of the network will not be identical to that of the ground-truth, it is in effect encountering a new situation at test-time. The acRNN structure compensates for this by linking the network's own predicted output into its future input streams during training, a similar approach to the technique proposed in (Bengio et al., 2015). Our method is light-weight and can be used in conjunction with any other RNN based learning scheme. Though straightforward, this technique fixes the issue of error accumulation, and allows the network to output incredibly long sequences without failure, on the order of hundreds of seconds (see Figure 5). Though we are yet as unable to prove the permanent stability of this structure, it seems empirically that motion can be generated without end. In summary, we present a new RNN training method capable for the first time of synthesizing potentially indefinitely long sequences of realistic and complex human motions with respect to different styles.

## 2  RELATED WORK

Many approaches have developed over the years in order to generate realistic human motion. In this section, we first review the literature that has dealt with motion synthesis using simulation methods and database-driven methods, then review the more recent deep learning approaches.

**Simulation-based Methods.**   Simulation-based techniques are able to produce physically plausible animations (Geijtenbeek & Pronost, 2012; Levine & Popović, 2012; Clegg et al., 2015; Hämäläinen et al., 2015; Ha et al., 2012; Liu et al., 2016), including realistic balancing, motion on terrain of various heights, and recovery from falling. Many of these methods consider physical constraints on the human skeleton while optimizing an motion objective. For example, in the work of Levine et al. (Levine & Popović, 2012), one task they employ is moving the skeleton in a certain direction without falling over. Similarly, in Ha et al. (Ha et al., 2012), given initial fall conditions, they seek to minimize joint stress due to landing impact while ensuring a desired landing pose. Though the output is plausible, they require specific objectives and constraints for each individual task; it is infeasible to apply such explicit priors to highly stylized and varied motion such as dancing or martial arts. There are also some recent works (Peng et al., 2017; Merel et al., 2017) which attempt to use less rigid objectives, instead employing adversarial and reinforcement learning. However, these motions often look uncanny and not human-like.

**Database-driven Methods.**   Motion graphs (Kovar et al., 2002; Safonova & Hodgins, 2007; Min & Chai, 2012), which stitch transitions into segments queried from a database, can generate locomotion along arbitrary paths, but are in essence limited to producing memorized sequences. Frameworks based on linear (Safonova et al., 2004; Chai & Hodgins, 2005; Tautges et al., 2011) and kernal methods (Mukai, 2011; Park et al., 2002; Levine et al., 2012; Grochow et al., 2004; Moeslund et al., 2006; Wang et al., 2008) have also shown reasonable success. Taylor et al. (Taylor & Hinton, 2009) use conditional restricted boltzmann machines to model motion styles. Grochow et al. (Grochow et al., 2004) and Wang et al. (Wang et al., 2005) both use Gaussian Process Latent Variable Models (GPLVM) to control motion. Levine et al. (Levine et al., 2012) apply reinforcement learning in a reduced space to compute optimal motions for a variety of motion tasks, including movement, punching, and kicking. Kee et al. (Lee et al., 2010) use a representation of motion data called motion-fields which allows users to interactively control movement of a character. The work of Xia

et al. (Xia et al., 2015) is able to achieve real-time style transfers of unlabeled motion. Taylor et al. (Taylor et al., 2007) use binary latent variables which are connected between different time steps while Liu et al. (Liu et al., 2005) estimate physical parameters from motion capture data. Database methods have limitations in synthesizing new variations of motion, in addition to the high memory costs for storing complex motions.

**Deep Learning Approaches.**   The use of recurrent networks is a natural approach to dealing with the problem of human motion. RNNs are trained to generate output sequentially, each output conditioned on the previous elements in the sequence. These networks have shown much success in Natural Language Processing for generating text (Sutskever et al., 2011), hand written characters (Graves, 2013; Gregor et al., 2015), and even captioning images (Vinyals et al., 2014). For the purpose of motion prediction and generation, Fragkiadaki et al. (Fragkiadaki et al., 2015) propose to use encoder-recurrent-decoder (ERD), jointly learning a skeleton embedding along with sequential information, in order to make body positions more realistic. In Jain et al. (Jain et al., 2016), the authors employ RNNs to learn spatio-temporal graphs of interaction, a structure which naturally applies to the human skeleton over time–joint positions are related over consecutive frames, and joints also interact with each other spatially (arms and legs interact with each other, the spine interacts with all other joints). Bütepage et al. (Bütepage et al., 2017) attempt to learn dance sequentially, but they are unable to produce varied and realistic output. More recently, Martinez et al. (Martinez et al., 2017) propose using a sequence-to-sequence architecture along with sampling-based loss. A main problem with these approaches, even in the case of the ERD, is that motion generation tends to converge to a mean position over time. Using our proposed training method, motion neither halts nor becomes unrecognizable for any period of time.

Holden et al. (Holden et al., 2015) demonstrate that a manifold of human motion can be learned using an autoencoder trained on the CMU dataset. They (Holden et al., 2016) extend this work by using this deep convolutional auto-encoder, in addition to a disambiguation network, in order to produce realistic walking motion along a user-defined path. They are also able to use the embedding space of the encoder to perform style transfer across different walks. Using the same autoencoder and a different disambiguation network, they can also perform other user defined tasks such as punching and kicking. Their method does not use an RNN structure and so it can only generate fixed length sequences based on the architecture of the disambiguation network. More recently, Holden et al. (Holden et al., 2017) use a Phase-Functioned Neural Network which takes the geometry of the scene into account to produce motion along a user-defined path. This method makes use of humans' periodic change of gait to synthesize realistic walking motion but has not demonstrated the ability to generate motions that have more complex step movements.

## 3    METHODOLOGY

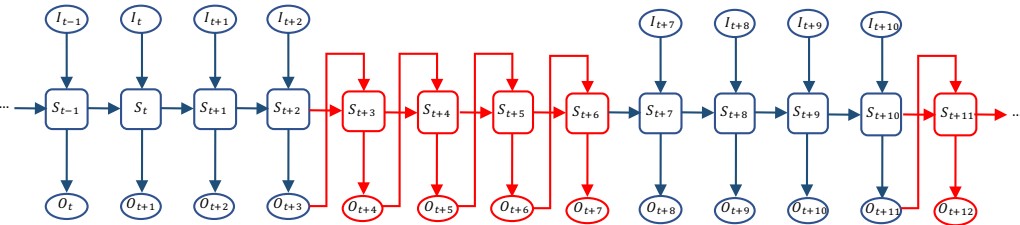

Figure 1: Visual diagram of an unrolled Auto-Conditioned RNN (right) with condition length $v = 4$ and ground truth length $u = 4$. $I_t$ is the input at time step $t$. $S_t$ is the hidden state. $O_t$ is the output.

**Auto-Conditioned RNN.**   Recurrent neural networks are well documented in the literature. A good survey and introduction can be found at (Karpathy, 2015; Olah, 2015). In short, a recurrent network can be represented as a function with a hidden memory unit, $x_{t+1} = f(x_t, m_t)$, where $m_t$ is the "memory" of the network, and is updated during every forward pass, and initialized at 0. The motivation is that the memory stores important information about a sequence up until that point, which can help with the prediction of the next element. In the experiments that follow, we use a

special type of RNN called an "LSTM", or a "long short term memory" network. We refer to the network trained with our method as "acLSTM".

As mentioned in the introduction, the major drawback of using LSTM/RNN deep learning methods for motion prediction is the problem of error accumulation. Following the conventional way to train an RNN, the network is recursively given a sequence of ground truth motion data, $G_{1,k} = [g_1, ..., g_k]$, and asked to produce the output $G_{2,k+1}[g_2, ..., g_{k+1}]$. Specifically, at training time, the recursive module at time step $t$, $M_t$, is conditioned on the input $[g_1, ..., g_{t-1}]$, and the error is measured between its output and $g_t$. Because of this, the backpropagation algorithm (Williams & Zipser, 1995) used for updating the parameters will always optimize w.r.t. the input ground-truth sequences $[g_1, ..., g_{t-1}]$. The parameters are accustomed to ground truth input — something it does not have access to at test-time. It is easy to see why problems will emerge: even if initial input is similar to ground truth, those slight differences will accumulate as the output is fed back in, producing output that become progressively worse until the sequence diverges or freezes. Effectively, the network is encountering a completely novel situation during test time as compared to training time, and so cannot perform well. The issue is so prevalent, in fact, that previous methods fail to produce realistic output after just several seconds (Fragkiadaki et al., 2015; Jain et al., 2016; Martinez et al., 2017).

Holden et al. (Holden et al., 2015) show that an autoencoder framework can to some degree "fix" such broken input, and some researchers have tried jointly learning such an encoder-decoder network alongside RNNs to better condition subsequent input (Gregor et al., 2015; Fragkiadaki et al., 2015). However, in the case of this framework being applied to motion as in ERD (Fragkiadaki et al., 2015), it does not generalize to indefinitely long sequences, as shown in (Jain et al., 2016). It seems as though the autoencoder might mitigate error accumulation, but does not eliminate it.

The acRNN, on the other hand, deals with poor network output explicitly by using it during training. Instead of only feeding in ground-truth instances, we use subsequences of the network's own outputs at periodic intervals. For instance, sticking with the example above, instead of conditioning the network on $G_{1,k} = [g_1, ..., g_k]$, we use $\hat{G}_{1,k} = [g_1, ..., g_u, p_{u+1}, ..., p_{u+v}, g_{u+v+1}.., g_k]$ to predict $G_{2,k+1} = [g_2, ..., g_{k+1}]$. The variable $p_{u+1}$ is the network output conditioned on $[g_1, ..., g_u]$, and $p_{u+2}$ is conditioned on $[g_1, ..., g_u, p_{u+1}]$. In this example, we refer to $v$ as the "condition length" and $u$ as the "ground-truth length". As the network is conditioned on its own output during training, it is able to deal with such input during synthesis. Figure 1 details an unrolled Auto-Conditioned RNN with condition length $u = v = 4$, and Figure 10 shows a more detailed view or our network. The method of (Bengio et al., 2015) also proposes using network output during training, but does so stochastically, without fixing condition lengths. However, we found that changing the condition/ground-truth length while keeping the proportion of ground-truth input fixed affects both the accuracy and variation of the output. See Figure 9 in the appendix.

Auto-conditioning also has the interpretation of training the network to produce longer sequences without further input. Whereas with standard training the network error is measured only against $p_{u+1}$ when conditioned with $[g_1, ..., g_u]$, under auto-conditioning the error is computed on the entire sequence $p_{u+1}, ..., p_{u+v}$ w.r.t. the same input. This effectively forces the network to produce $v$ frames of output simultaneously as opposed to only one. Martinez et al. (Martinez et al., 2017) also use contiguous sequences of network output during training, but unlike us they do not alternate these with ground-truth input at regular intervals.

**Data Representation.** We use the publicly available CMU motion-capture dataset for our experiments. The dataset is given as sequences of 57 skeleton joint positions in 3D-space. First, we define a root joint, whose position at time $t$ is given by $r_t = (r_{1,t}, r_{2,t}, r_{3,t})$. In order to better capture relative motion, we instead use the displacement of the root from the previous frame for input — $\tilde{r}_t = (r_{1,t} - r_{1,t-1}, r_{2,t} - r_{2,t-1}, r_{3,t} - r_{3,t-1})$. For every other joint at time $t$, with position $j_t = (j_{1,t}, j_{2,t}, j_{3,t})$, we represent it as as the relative distance in the world-coordinate system to the root joint, $\tilde{j}_t = (j_{1,t} - r_{1,t}, j_{2,t} - r_{2,t}, j_{3,t} - r_{3,t})$. All distances are stored in meters. We use a skeleton with height 1.54 meters in neutral pose for all experiments.

We found this representation to be desirable for several reasons. Primarily, if there is periodic motion in the dataset, we would like frames at the same point in the repeated activity to have small Euclidean distance. If we instead used absolute positions, even if it were only for the hip, this would certainly not be the case. We note that there are alternative representations which achieve the same property. (Fragkiadaki et al., 2015) express joint positions as rotations relative to a parent joint, and (Holden

et al., 2016) define them in the body's relative coordinate system along with a relative rotation of the body w.r.t. the previous frame.

**Training.** We train the acLSTM with three fully connected layers with a memory size of 1024, similar to (Fragkiadaki et al., 2015; Jain et al., 2016; Bütepage et al., 2017). The main difference is that for every $u$ ground-truth inputs of the time series, we connect $v$ instances of the network's own output into its subsequent input streams (see section 2.1). In the main body of the paper, we set $u = v = 5$. We carry out further experiments with varying $u$ and $v$ in the appendix. We train with a sequence length of 100 for 500000 iterations using the ADAM backpropagation algorithm (Kingma & Ba, 2014) on an NVIDIA 1080 GPU for each dataset we experiment on. We use Euclidean loss for our objective function. The initial learning rate is is set to 0.0001. We implement the training using the python caffe framework (Jia et al., 2014). We sample sequences at multiple frame-rates as well as rotate the sequence randomly in order to increase the training size.

In detail, if at time $t$ we input the network H with ground truth, then the loss is given by:

$$\mathrm{L}(x_t) = ||\mathrm{H}(x_t, m_t) - x_{t+1}||_2^2 \tag{1}$$

where $x_t$ and $x_{t+1}$ are the ground truth motions for time steps $t$ and $t + 1$.

If at time $t$ the network is input with its own previous output, the loss is given by:

$$\mathrm{L}(x_t) = \left\|\mathrm{H}(\hat{x_t}^k, m_t) - x_{t+1}\right\|_2^2 \tag{2}$$

where $\hat{x_t}^k = \mathrm{H}(\mathrm{H}(...\mathrm{H}(x_{t-k}, m_{t-k}), m_{t-1}), m_t)$. $k$ indicates how many times the network has fed itself is its own input since the last injection of ground-truth. It is bounded by the condition length (see previous section).

## 4 EXPERIMENTAL RESULTS

We evaluate our synthesized motion results on different networks trained on each of four distinct subsets from the CMU motion capture database: martial arts, Indian dance, Indian/salsa hybrid, and walking. An anonymous video of the results can be found here: https://youtu.be/FunMxjmDIQM.

**Quantitative Results.** Table 1 shows the prediction error as Euclidean distance from the ground truth for different motion styles at various time frames.

We compare with a 3-layer LSTM (LSTM-3LR), ERD (Fragkiadaki et al., 2015), the seq2seq framework of (Martinez et al., 2017) as well as scheduled sampling (Bengio et al., 2015). Though our goal is long-term stable synthesis and not prediction, it can be seen in Table 1 that acLSTM performs reasonably well in this regard, even achieving the best performance at most time-scales for the Indian dance category. The method of (Martinez et al., 2017) performs the best in the short term, but has the worst error by far past the half-second mark. As noted in (Fragkiadaki et al., 2015), the stochasticity of human motion makes longer-term prediction infeasible. Table 2 in the appendix shows this error difference between versions of the Indian dance network trained with different condition lengths. Figure 2 shows the average change between subsequent frames for different frame times for the acLSTM and the basic scheme. For the more complex motions of Martial Arts and Indian Dance, it is clear from the graph that acLSTM continues producing motion in the long-term while the basic training scheme (LSTM-3LR) without auto-conditioning results in stagnated motion, as the network freezes into a converged mean position. Likewise, Figure 9 in the appendix shows this average change for the Indian dance network trained with different condition lengths. We note that while the methods of (Fragkiadaki et al., 2015; Martinez et al., 2017) do not simply freeze completely, as with the basic scheme, their motion becomes unrealistic at around the same time (Figure 7). This is consistent with the observations of the original authors.

**Qualitative Results.** Figure 4 shows several example frames taken from 50 second synthesized outputs, representing both the extended long term complexity and plausibility of the output. In comparison, our implementations of ERD, and seq2seq are only able to generate motion for a couple

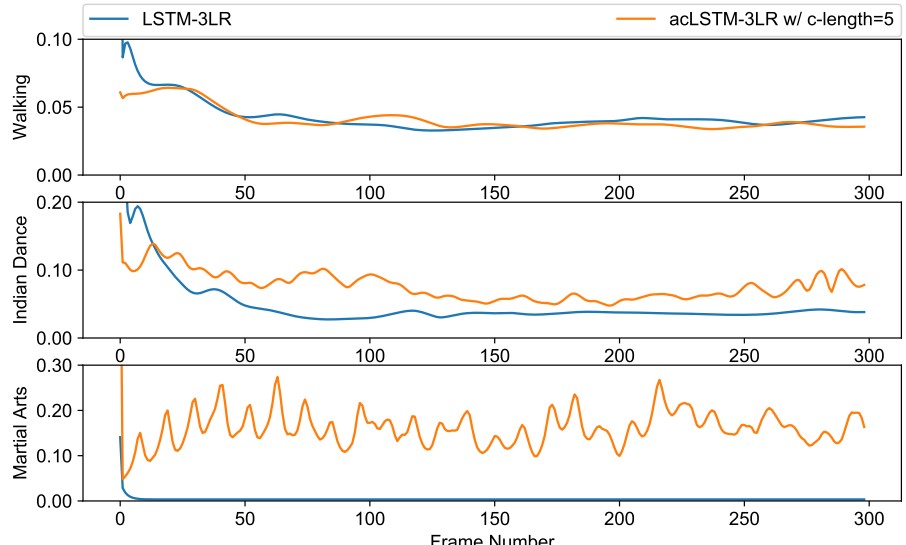

Figure 2: **Motion change** between subsequent frames of different motion styles, given as Euclidean distance in prediction results, at different frames. All acLSTM networks here are trained with condition length 5. Predictions are generated with 10 frames (approximately 170 ms) of seed motion from test set. Results are averaged over 20 random seed motions. Low value in motion change indicates the freezing of motion. Note that acLSTM and vanilla have exactly the same architecture - differences are due solely to training. Results averaged over 20 seed motions.

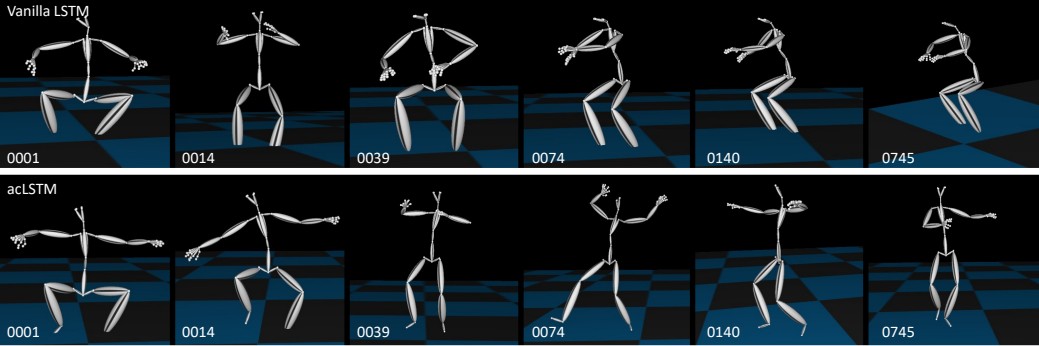

Figure 3: Comparison between the vanilla LSTM and our method at 250,000 iterations of training. **top:** vanilla LSTM, **bottom:** acLSTM. The two synthesized motions are initialized with the same 10 frames of ground truth motion. The motion generated by the vanilla LSTM freezes after around 60 frames. Our method does not freeze.

of seconds before they become unrealistic (Figure 7). Scheduled sampling (Bengio et al., 2015) performs better than ERD and seq2seq, but also freezes eventually, perhaps because it does not force the network to generate consistently longer sequences during training as our method does. We also demonstrate the possibility of creating hybrid motions by mixing training sets in third row of Figure 4.

It should be noted that the motion in our framework, while never permanently failing, also does not remain perfectly realistic, and short-term freezing does sometimes occur. This, however, does not occur for the network trained just on walking. It is perhaps that the movement of the feet in conjunction with the absolute movement of the root joint is not as easy to discern when the feet leave the ground aperiodically, or there are jumping motions.

Table 1: **Motion prediction error** for different styles of motion at {80, 160, 240, 320, 400, 480, 560, 640} ms after seed motion of 10 frames (approximately 170 ms) from test set. All acLSTM networks here are trained with condition length 5. Error given as Euclidean distance from the ground truth for the corresponding frame. All results averaged over 20 random seed motions. Longer motion prediction is not feasible due to randomness of human motion.

| Architecture | 80 ms | 160 ms | 240 ms | 320 ms | 400 ms | 480 ms | 560 ms | 640 ms |
|---|---|---|---|---|---|---|---|---|
| Walking | | | | | | | | |
| LSTM-3LR | 2.46 | 2.41 | 2.28 | 2.27 | 2.41 | 2.65 | 2.91 | 3.15 |
| acLSTM-3LR | 1.05 | 1.77 | 2.20 | 2.46 | 2.66 | 2.79 | 2.99 | 3.24 |
| ERD (Fragkiadaki et al., 2015) | 0.13 | 0.22 | 0.34 | 0.50 | **0.71** | **0.95** | 1.19 | 1.42 |
| seq2seq (Martinez et al., 2017) | **0.09** | **0.13** | **0.24** | **0.42** | 0.74 | 1.22 | 1.85 | 2.79 |
| sch. smp. (Bengio et al., 2015) | 0.42 | 0.56 | 0.71 | 0.83 | 0.93 | 0.99 | **1.02** | **1.05** |
| Indian Dance | | | | | | | | |
| LSTM-3LR | 2.82 | 2.83 | 2.85 | 2.88 | 2.89 | 2.90 | 2.92 | 2.92 |
| acLSTM-3LR | 0.685 | 0.99 | **1.22** | **1.53** | **1.89** | **2.08** | **2.27** | **2.55** |
| ERD (Fragkiadaki et al., 2015) | 0.51 | **0.74** | 1.25 | 1.96 | 2.62 | 3.31 | 3.76 | 3.86 |
| seq2seq (Martinez et al., 2017) | **0.49** | 0.79 | 1.48 | 2.95 | 5.41 | 8.88 | 13.29 | 18.73 |
| sch. smp. (Bengio et al., 2015) | 1.54 | 2.24 | 2.49 | 2.52 | 2.65 | 2.90 | 2.94 | 3.12 |
| Martial Arts | | | | | | | | |
| LSTM-3LR | 0.69 | 0.86 | 1.11 | 1.36 | 1.60 | 1.83 | 2.03 | 2.19 |
| acLSTM-3LR | 0.52 | 0.74 | 0.95 | 1.14 | 1.35 | 1.56 | 1.73 | 1.88 |
| ERD (Fragkiadaki et al., 2015) | 0.32 | 0.44 | **0.63** | **0.90** | 1.14 | 1.40 | 1.61 | 1.88 |
| seq2seq (Martinez et al., 2017) | **0.28** | **0.43** | 0.87 | 1.57 | 2.53 | 3.89 | 5.83 | 8.62 |
| sch. smp. (Bengio et al., 2015) | 0.63 | 0.86 | 0.91 | 0.98 | **1.07** | **1.12** | **1.20** | **1.28** |

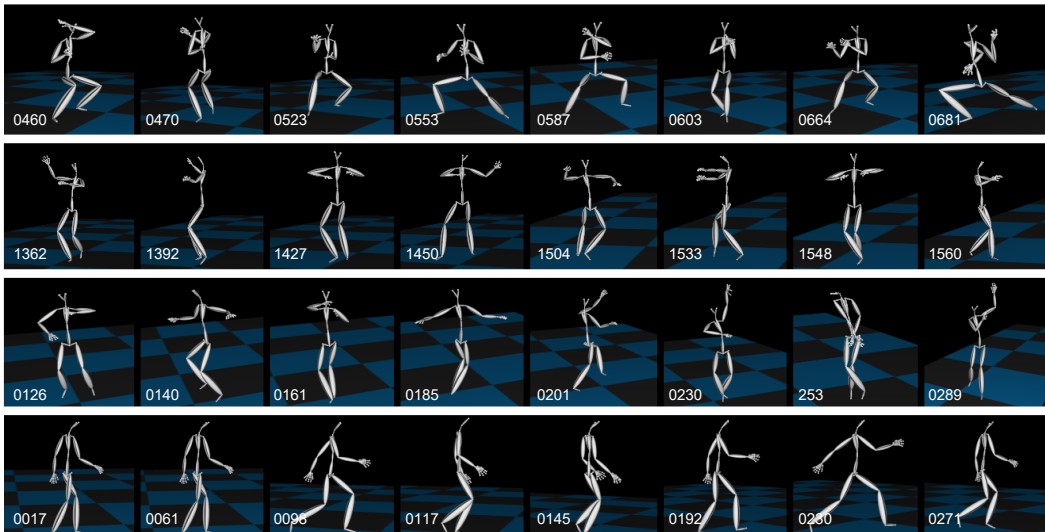

Figure 4: Motion sequences generated by acLSTM, sampled at various frames. Motion style from **top to bottom**: martial arts, Indian dancing, Indian/salsa hybrid and walking. All the motions are generated at 60 fps, and are initialized with 10 frames of ground truth data randomly picked up from the database. The number at the bottom of each image is the frame index. The images are rendered with BVHViewer 1.1 (van Basten, 2017)

When motion does stagnate, it recovers relatively quickly, and the motion never diverges or freezes completely (see Figure 6). The short-term freezing could possibly be explained by "dead-times" in the training sequence, where the actor is beginning or ending a sequence which involves a rest position. Note that when training with two different datasets, as in the case of the Indian/Salsa combined network, motion borrows from both dance styles. We also demonstrate in Figure 5 that our method does not freeze even after 20,000 frames of synthesis, which is approximately 333 seconds of output.

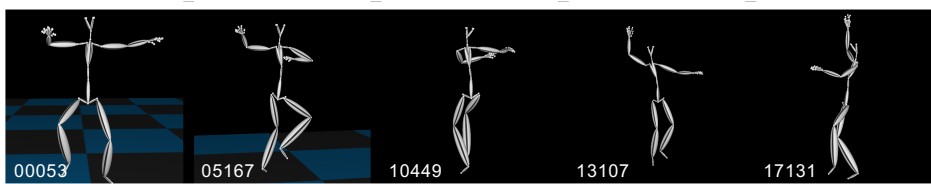

Figure 5: Sample frames from a 300+ second generated sequence. Note that no sequence in the training set exceeds 30 seconds of contiguous motion.

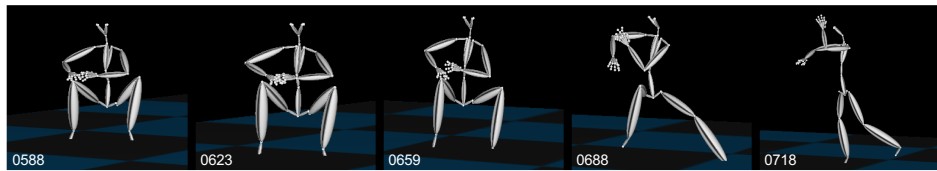

Figure 6: Example of the acLSTM recovering from short term stagnated motion.

One can see a qualitative comparison of acLSTM with a basic LSTM-3LR in Figure 3, both trained on the Indian dance dataset. We find the performance of the vanilla network to be consistent with the results reported in (Fragkiadaki et al., 2015; Jain et al., 2016; Bütepage et al., 2017; Martinez et al., 2017), freezing at around 1000 ms. It never recovers from the motion. Our network, on the other hand, continues producing varied motion for the same time frame.

## 5 DISCUSSION AND FUTURE WORK

We have shown the effectiveness of the acLSTM architecture to produce extended sequences of complex human motion. We believe our work demonstrates qualitative state-of-the-art results in motion generation, as all previous work has focused on synthesizing relatively simple human motion for extremely short time periods. These works demonstrate motion generation up to a couple of seconds at most while acLSTM does not fail even after over 300 seconds. Though we are as of yet unable to prove indefinite stability, it seems empirically that acLSTM can generate arbitrarily long sequences. Current problems that exist include choppy motion at times, self-collision of the skeleton, and unrealistic sliding of the feet. Further developement of GAN methods, such as (Lamb et al., 2016), could result in increased realism, though these models are notoriously hard to train as they often result in mode collapse. Combining our technique with physically based simulation to ensure realism after synthesis is also a potential next step. Finally, it is important to study the effects of using various condition lengths during training. We begin the exploration of this topic in the appendix, but further analysis is needed.

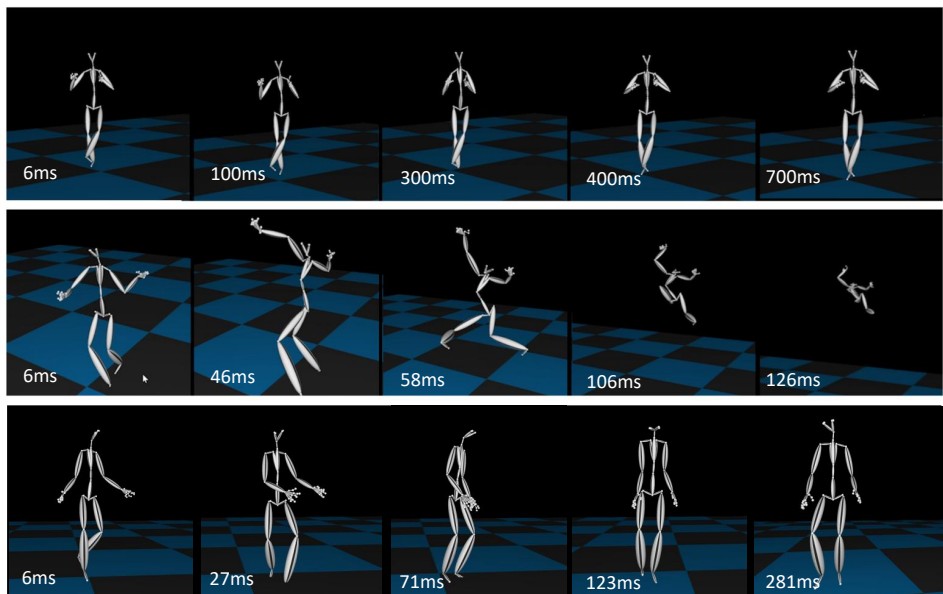

Figure 7: Frames of alternative methods at latest failure-points out of 20 generated 1000-frame motion samples. acLSTM did not fail in any of 20 generated samples. **Top**: seq2seq failure starting around frame 40. Motion becomes unrealistic and does not recover. **Middle**: ERD failure between frames 400 and 500; slight motion occurs, but pose remains unaltered. **Bottom**: scheduled sampling failure. Examples trained on Indian Dance.

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

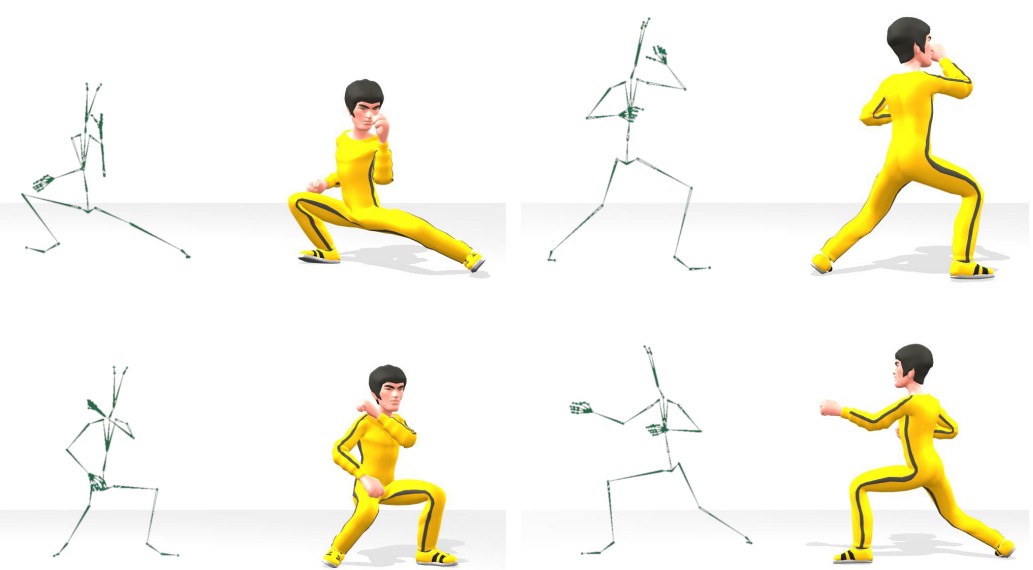

Figure 8: Selected frames of a rigged animation example using the martial arts network.

## A  THE EFFECT OF CONDITION LENGTH

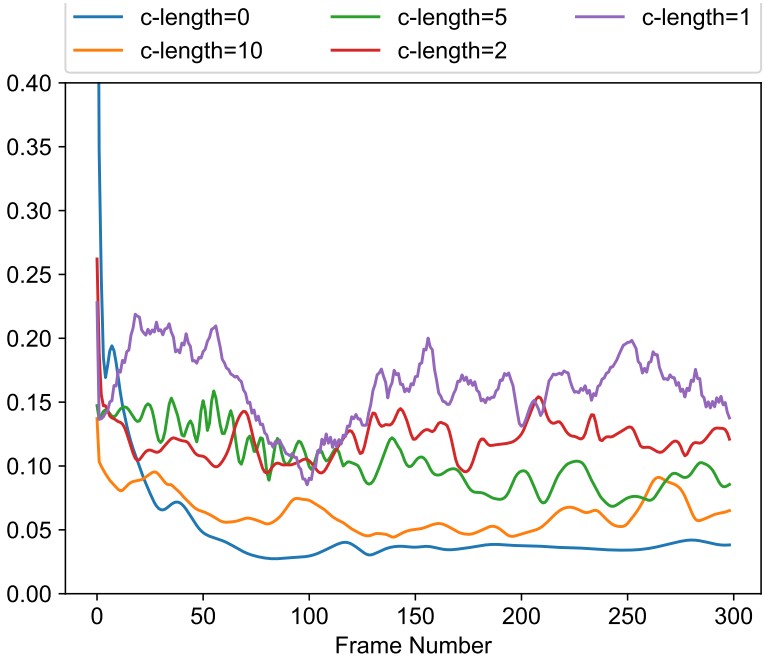

Figure 9: **Motion change** between subsequent frames using different condition lengths, given as Euclidean distance in prediction results, at different frames. Predictions are initialized with 10 frames (approximately 170 ms) of seed motion from test set. Results are averaged over 20 random seed motions. All networks are trained on the Indian dance dataset.

Table 2: **Motion prediction error** for different condition lengths {80, 160, 240, 320, 400, 480, 560, 640} ms after seed motion of 10 frames (approximately 170 ms) from test set. Error given as Euclidean distance from the ground truth for the corresponding frame. All results averaged over 20 random seed motions. All networks are trained on the Indian dance dataset

| Architecture | 80 ms | 160 ms | 240 ms | 320 ms | 400 ms | 480 ms | 560 ms | 640 ms |
|---|---|---|---|---|---|---|---|---|
| Indian Dance | | | | | | | | |
| c-length 10 | **0.41** | **0.52** | **0.68** | **0.86** | **1.04** | **1.23** | **1.39** | **1.55** |
| c-length 5 | 0.685 | 0.99 | 1.22 | 1.53 | 1.89 | 2.08 | 2.27 | 2.55 |
| c-length 2 | 0.83 | 1.28 | 1.61 | 1.87 | 2.09 | 2.31 | 2.51 | 2.66 |
| c-length 1 | 0.89 | 1.40 | 1.78 | 2.14 | 2.48 | 2.70 | 2.84 | 2.93 |
| sch. smp. | 1.54 | 2.24 | 2.49 | 2.52 | 2.65 | 2.90 | 2.94 | 3.12 |

# B  PREDICTION ERROR

It seems that Table 2 and Figure 9 might imply some sort of trade off between motion change over time and short-term motion prediction error when training with different condition lengths. However, it is also possible that limiting motion magnitude on this particular dataset might correspond to lower error. Further experiments of various condition lengths on several motion styles need to be conducted to say anything meaningful about the effect.

# C  VISUAL DIAGRAM OF AUTO-CONDITIONED LSTM

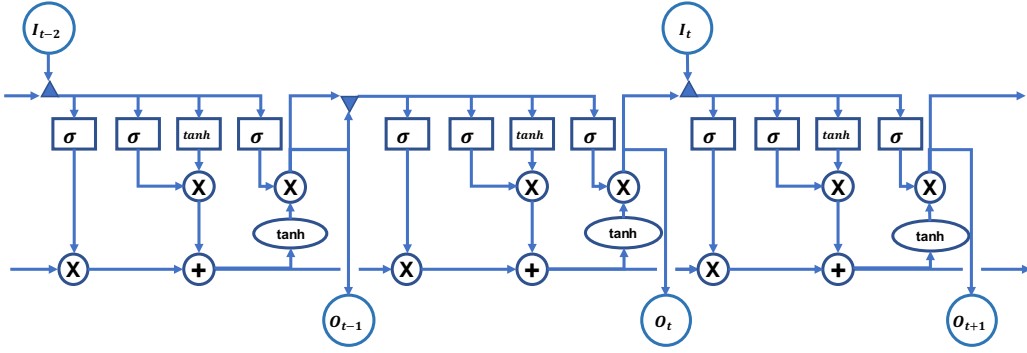

Figure 10: Detailed visual diagram of an unrolled Auto-Conditioned LSTM. $I_t$ is the input at time step $t$. $O_t$ is the output state. Rectangles indicate the neural network layer. Circles indicate point wise operation. Triangles indicate concatenation.

