# OpenReview forum: "Auto-Conditioned Recurrent Networks for Extended Complex Human Motion Synthesis"
_ICLR.cc/2018/Conference — Accept (Poster)_

### Official Review · AnonReviewer1 · 2017-11-25
**A useful technique to avoid prediction error accumulation**

**Rating:** 7
**Confidence:** 3

**Review:**

This paper proposes acLSTM to synthesize long sequences of human motion. It tackles the challenge of error accumulation of traditional techniques to predict long sequences step by step. The key idea is to combine prediction and ground truth in training. It is impressive that this architecture can predict hundreds of frames without major artifacts.

The exposition is mostly clear. My only suggestion is to use either time (seconds) or frame number consistently. In the text, the paper sometimes use time, and other time uses frame index (e.g. figure 7 and its caption). It confuses me a bit since it is not immediate clear what the frame rate is.

In evaluation, I think that it is important to analyze the effect of condition length in the main text, not in the Appendix. To me, this is the most important quantitive evaluation that give me the insight of acLSTM. It also gives a practical guidance to readers how to tune the condition length. As indicated in Appendix B, "Further experiments need to be conducted to say anything meaningful." I really hope that in the next version of this paper, a detailed analysis about condition length could be added.

In summary, I like the method proposed in the paper. The result is impressive. I have not seen an LSTM based architecture predicting a complex motion sequence for that long. However, more detailed analysis about condition length is needed to make this paper complete and more valuable.

---

### Official Review · AnonReviewer3 · 2017-11-26
**Review:   generally positive; some reservations**

**Rating:** 7
**Confidence:** 5

**Review:**

The problem of learning auto-regressive (data-driven) human motion models that have long-term stability
is of ongoing interest. Steady progress is being made on this problem, and this paper adds to that.
The paper is clearly written. The specific form of training (a fixed number of self-conditioned predictions,
followed by a fixed number of ground-truth conditioned steps) is interesting for simplicity and its efficacy.
The biggest open question for me is how it would compare to the equally simple stochastic version proposed
by the scheduled sampling approach of [Bengio et al. 2015].

PROS:  The paper provides a simple solution to a problem of interest to many.
CONS:  It is not clear if it improves over something like scheduled sampling, which is a stochastic predecessor
       of the main idea introduced here. The "duration of stability" is a less interesting goal than
       actually matching the distribution of the input data.

The need to pay attention to the distribution-mismatch problem for sequence prediction problems
has been known for a while. In particular, the DAGGER (see below) and scheduled sampling algorithms (already cited)
target this issue, in addition to the addition of progressively increasing amounts of noise during training
(Fragkiadaki et al). Also see papers below on Professor Forcing, as well as "Learning Human Motion Models
for Long-term Predictions" (concurrent work?), which uses annealing over dropout rates to achieve stable long-term predictions.

  DAGGER algorithm (2011):  http://www.jmlr.org/proceedings/papers/v15/ross11a/ross11a.pdf
  "A Reduction of Imitation Learning and Structured Prediction to No-Regret Online Learning"

  Professor Forcing (NIPS 2016)
  http://papers.nips.cc/paper/6099-professor-forcing-a-new-algorithm-for-training-recurrent-networks.pdf

  Learning Human Motion Models for Long-term Predictions (2017)
  https://arxiv.org/abs/1704.02827
  https://www.youtube.com/watch?v=PgJ2kZR9V5w

While the motions do not freeze, do the synthesized motion distributions match the actual data distributions?
This is not clear, and would be relatively simple to evaluate.  Is the motion generation fully deterministic?
It would be useful to have probabilistic transition distributions that match those seen in the data.
An interesting open issue (in motion, but also of course NLP domains) is that of how to best evaulate
sequence-prediction models.  The duration of "stable prediction" does not directly capture the motion quality.

Figure 1:  Suggest to make u != v for the purposes of clarity, so that they can be more easily distinguished.

Data representation:
Why not factor out the facing angle, i.e., rotation about the vertical axis, as done by Holden et al, and in a variety of
previous work in general?
The representation is already made translation invariant. Relatedly, in the Training section,
data augmentation includes translating the sequence: "rotate and translate the sequence randomly".
Why bother with the translation if the representation itself is already translation invariant?

The video illustrates motions with and without "foot alignment".
However, no motivation or description of "foot alignment" is given in the paper.

The following comment need not be given much weight in terms of evaluation of the paper, given that the
current paper does not use simulation-based methods. However, it is included for completeness.
The survey of simulation-based methods for modeling human motions is not representative of the body of work in this area
over the past 25 years.  It may be more useful to reference a survey, such as
"Interactive Character Animation Using Simulated Physics: A State‐of‐the‐Art Review" (2012)
An example of recent SOTA work for modeling dynamic motions from motion capture, including many
highly dynamic motions, is "Guided Learning of Control Graphs for Physics-Based Characters" (2016)
More recent work includes "Learning human behaviors from motion capture by adversarial imitation",
"Robust Imitation of Diverse Behaviors", and "Deeploco: Dynamic locomotion skills using hierarchical deep reinforcement learning", all of which demonstrate imitation of various motion styles to various degrees.

It is worthwhile acknowledging that the synthesized motions are still low quality, particular when rendered with more human-like looking models, and readily distinguishable from the original motions.  In this sense, they are not comparable to the quality of results demonstrated in recent works by Holden et al. or some other recent works.  However, the authors should be given credit for including some results with fully rendered characters, which much more readily exposes motion flaws.

The followup work on [Lee et al 2010 "Motion Fields"] is quite relevant:
"Continuous character control with low-dimensional embeddings"
In terms of usefulness, being able to provide some control over the motion output is a more interesting problem than
being able to generate long uncontrolled sequences.  A caveat is that the methods are not applied to large datasets.

---

### Official Review · AnonReviewer2 · 2017-11-27
**Well written paper with somewhat limited novelty but state-of-the-art results**

**Rating:** 6
**Confidence:** 5

**Review:**

Paper presents an approach for conditional human (skeleton) motion generation using a form of the LSTM, called auto-conditioned LSTM (acLSTM). The key difference of acLSTM is that in it parts of the generated sequences, at regular intervals, are conditioned on generated data (as opposed to just ground truth data). In this way, it is claimed that acLSTM can anticipate and correct wrong predictions better than traditional LSTM models that only condition generation on ground truth when training. It is shown that trained models are more accurate at long-term prediction (while being a bit less accurate in short-term prediction).

Generally the idea is very sensible. The novelty is somewhat small, given the fact that a number of other methods have been proposed to address the explored challenge in other domains. The cited paper by Bengio et al., 2015 is among such, but by no means the only one. For example, “Professor Forcing: A New Algorithm  for Training Recurrent Nets” by Goyal et al. is a more recent variant that does away with the bias that the scheduled sampling of Bengio et al., 2015 would introduce. The lack of comparison to these different methods of training RNNs/LSTMs with generated or mixture of ground truth and generated data is the biggest shortcoming of the paper. That said, the results appear to be quite good in practice, as compared to other state-of-the-art methods that do not use such methods to train.

Other comments and corrections:

- The discussion about the issues addressed not arising in NLP is in fact wrong. These issues are prevalent in training of any RNN/LSTM model. In particular, similar approaches have been used in the latest image captioning literature.

- In the text, when describing Figure 1, unrolling of u=v=1 is mentioned. This is incorrect; u=v=4 in the figure.

- Daniel Holden reference should not contain et. al. (page 9)

---

### Author Response · Authors · 2017-12-24
**Response to Reviews**

We would like to thank all the reviewers. We especially appreciate being informed of relevant works which we have overlooked and mistakes in the paper, which we are happy to add/revise. However, one work mentioned by Reviewer3, "Learning Human Motion Models for Long-term Predictions (2017)", is not currently peer-reviewed and so we do not feel a need for its inclusion.  Furthermore, the proposed approach therein is similar to "Recurrent Network Models for Human Dynamics", which we already compare with.

Both Rev2 and Rev3 suggest including scheduled sampling and professor forcing for comparison. We agree that adding a comparison with scheduled sampling is definitely appropriate and it will be included in the final upload. Regarding professor forcing, no publicly available implementation currently exists, and we could not get a working implementation even after contacting the authors. We are currently working on our own implementation of professor forcing, but we cannot seem to make it work.  GANs are notoriously finicky to work with and require a lot of hyperparameter turning, so this result is not unexpected.

We further note that no GAN approaches have to date been shown effective for the problem of human-motion generation.  Given this, and our initial experiments with it, it is our feeling that successfully using GAN approaches for generating human motion is in and of itself a noteworthy research problem, and exploration of the effects of professor-forcing should be addressed in such a work, but is outside the scope of this paper.

Q (R3): What is "foot alignment"?
A: We postprocess the animation so that when the foot is in contact with the ground (the height of the foot is close to 0), any motion of the foot in the XZ plane is stopped, and the rest of the body moves instead. This is an easy fix to "sliding" feet in the animation, while keeping the relative pose of the skeleton the same.

Q (R3): Why not factor out the facing angle, i.e., rotation about the vertical axis, as done by Holden et al, and in a variety of
previous work in general?
A: No particular reason - we simply decided we wanted to keep the prediction format consistent.  If we factored out the facing angle, then we would need to predict relative hip-rotation per-frame in addition to displacement.  We decided to predict only displacement.

Q: (R3): Why bother with the translation if the representation itself is already translation invariant?
A: Thanks for pointing this out.  In earlier experiments, we predicted the absolute hip position at every frame, instead of its relative displacement from the hip of the previous frame. You are correct - in the current formulation, it is translation invariant.  We will edit the text to reflect that.

Q: (R3): Is the motion generation fully deterministic?
A: Yes. There is no probabilistic model involved.  Each frame of motion is completely determined by a 171-dimensional vector, representing the positions of 57 joint locations.  We predict these joint locations in space, using L2 norm during training.

Q: (R3): An interesting open issue (in motion, but also of course NLP domains) is that of how to best evaulate
sequence-prediction models.  The duration of "stable prediction" does not directly capture the motion quality.
A: This is definitely true, and it is apparent at times that our motion is not realistic. The reason we focused much of the discussion on duration is because previous works were unable to achieve even this, and duration is clearly a precondition necessary for further evaluation of quality. Previous works were not able to generate stable motion for more than a couple of seconds for simple motions such as walking or smoking, let alone dancing.  Without first establishing a method that can at least run for a reasonable amount of time without failing, serious discussion of motion quality is impossible.

Q: (R3): While the motions do not freeze, do the synthesized motion distributions match the actual data distributions?
This is not clear, and would be relatively simple to evaluate.
A: It seems that the networks trained on distinct datasets reflect features unique to those datasets: martial arts motion shows punching, kicking; dancing networks shows dance, and the walking/running network only outputs continuous walking/running. Do you have in mind additional quantitative evaluations for comparing distributions, besides euclidean error?  We are happy to consider it.

Q (R1): I really hope that in the next version of this paper, a detailed analysis about condition length could be added.
A: We agree this would be ideal, but to fully address this issue, further theoretical analysis is also necessary.  We are currently working on providing such theoretical work (WHY auto-conditioning works), which we believe is appropriate for future work. There is not much to say about the quantitative results on condition-length currently, which is why they are included in the appendix.

---

### Decision · Program_Chairs · 2018-01-29
**ICLR 2018 Conference Acceptance Decision**

**Decision:**

Accept (Poster)

**Comment:**

This paper proposes a real-time method for synthesizing human motion of highly complex styles. The key concern raised by R2 was that the method did not depart greatly from a standard LSTM: parts of the generated sequences are conditioned on generated data as opposed to ground truth data. However, the reviewer thought the idea was sensible and the results were very good in practice. R1 also agreed that the results were very good and asked for a more detailed analysis of conditioning length and some clarification. R3 brought up similarities to Professor Forcing (Goyal et al. 2016) -- also noted by R2 -- and Learning Human Motion Models for Long-term Predictions (Ghosh et al. 2017) -- noting not peer-reviewed. R3 also raised the open issue of how to best evaluate sequence prediction models like these. They brought up an interesting point, which was that the synthesized motions were low quality compared to recent works by Holden et al., however, they acknowledged that by rendering the characters this exposed the motion flaws. The authors responded to all of the reviews, committing to a comparison to Scheduled Sampling, though a comparison to Professor Forcing was proving difficult in the review timeline. While this paper may not receive the highest novelty score, I agree with the reviewers that it has merit. It is well written, has clear and reasonably thorough experiments, and the results are indeed good.